# Metabolic reprogramming during neuronal differentiation from aerobic glycolysis to neuronal oxidative phosphorylation

Xinde Zheng[1†], Leah Boyer[2†], Mingji Jin[1], Jerome Mertens[2], Yongsung Kim[2], Li Ma[1,3], Li Ma[1,3], Michael Hamm[1], Fred H Gage[2*], Tony Hunter[1*]

[1]Molecular and Cell Biology Laboratory, Salk Institute for Biological Studies, La Jolla, United States; [2]Laboratory of Genetics, Salk Institute, La Jolla, United States; [3]Gene Expression Laboratory, Salk Institute, La Jolla, United States

*For correspondence: gage@salk.edu (FHG); hunter@salk.edu (TH)

†These authors contributed equally to this work

Competing interests: The authors declare that no competing interests exist.

**Abstract** How metabolism is reprogrammed during neuronal differentiation is unknown. We found that the loss of hexokinase (HK2) and lactate dehydrogenase (LDHA) expression, together with a switch in pyruvate kinase gene splicing from PKM2 to PKM1, marks the transition from aerobic glycolysis in neural progenitor cells (NPC) to neuronal oxidative phosphorylation. The protein levels of c-MYC and N-MYC, transcriptional activators of the HK2 and LDHA genes, decrease dramatically. Constitutive expression of HK2 and LDHA during differentiation leads to neuronal cell death, indicating that the shut-off aerobic glycolysis is essential for neuronal survival. The metabolic regulators PGC-1α and ERRγ increase significantly upon neuronal differentiation to sustain the transcription of metabolic and mitochondrial genes, whose levels are unchanged compared to NPCs, revealing distinct transcriptional regulation of metabolic genes in the proliferation and post-mitotic differentiation states. Mitochondrial mass increases proportionally with neuronal mass growth, indicating an unknown mechanism linking mitochondrial biogenesis to cell size.

## Introduction

Neurons rely on oxidative phosphorylation to meet energy demands. Malfunctions of mitochondrial oxidative phosphorylation (OXPHOS) lead to a wide range of neurological disorders, and are frequently observed in neurodegenerative diseases (*Lin and Beal, 2006*; *Schon and Przedborski, 2011*; *Koopman et al., 2013*). For example, chronic exposure of the brain to the lipophilic pesticide rotenone causes dopaminergic neuron degeneration (*Betarbet et al., 2000*). Hereditary mutations in OXPHOS genes cause Leigh syndrome, a severe early childhood neurodegeneration (*Finsterer, 2008*). Little is known about the molecular basis of neurons' preference for oxidative phosphorylation, or neuronal metabolic responses to the energy crisis and their implications for disease progression. It is imperative to understand how neuronal metabolism, defined by the metabolic enzymes in pathways, such as glycolysis, TCA cycle and mitochondrial OXPHOS, is set up during development, is maintained during adult life and is altered in neurological disorders.

Glycolysis and TCA cycle are major pathways providing metabolic precursors for biosynthesis and energy production. The activities and metabolic flux of these pathways are delicately tuned to ensure optimal resource distribution, conforming to cellular function. The balance between glycolysis and mitochondrial oxidative phosphorylation is essential for stem cell function (*Shyh-Chang et al., 2013*; *Teslaa and Teitell, 2015*). For instance, deletion of lactate dehydrogenase A (LDHA) impairs

**eLife digest** Structures called mitochondria act like the batteries of cells, and use several different metabolic processes to release energy. For example, neurons rely on a metabolic process called oxidative phosphorylation, while neural progenitor cells (which develop, or differentiate, into neurons) use a process called aerobic glycolysis instead. Little is known about why neurons prefer to use oxidative phosphorylation to provide them with energy, and it is also not clear why problems that affect this process are often seen in neurological disorders and neurodegenerative diseases.

Zheng, Boyer et al. have now used human neural progenitor cells to explore the metabolic changes that occur as these cells develop into neurons. It appears that the loss of two metabolic enzymes, called hexokinase and lactate dehydrogenase, marks the transition from aerobic glycolysis to oxidative phosphorylation. In addition, the instructions to produce an enzyme called pyruvate kinase are altered or "alternatively spliced" when progenitor cells differentiate, which in turn changes the structure of the enzyme. The levels of the proteins that activate and regulate the production of these three metabolic enzymes also decrease dramatically during this transition. Further experiments showed that neurons that produce hexokinase and lactate dehydrogenase while they differentiate die, which means that neurons must shut off aerobic glycolysis in order to survive.

The amounts of two proteins that regulate metabolism (called PGC-1α and ERRγ) increase significantly when a neuron differentiates. This sustains a constant level of activity for several metabolic and mitochondrial genes as neural progenitor cells differentiate to form neurons. Zheng, Boyer et al. also found that neurons build more mitochondria as they grow; this suggests that an unknown mechanism exists that links the creation of mitochondria to the size of the neuron.

Zheng, Boyer et al. have mainly focused on how much of each metabolic enzyme is produced inside cells, but these levels may not completely reflect the actual level of enzyme activity. The next steps are therefore to investigate whether any other processes or modifications play a part in regulating the enzymes. Further investigation is also needed to determine the effects of changes in mitochondrial structure that occur as a neuron develops from a neural progenitor cell.

maintenance and proliferation of hematopoietic stem and progenitor cells (*Wang et al., 2014*). Most cancer cells use aerobic glycolysis to generate energy, a phenomenon called the Warburg effect, in which a large fraction of glycolytic pyruvate is converted into lactate (*Vander Heiden et al., 2009*). Moreover, under low oxygen condition, the transcription of glycolytic genes is upregulated by hypoxia-inducible factor (HIF), and increased glycolytic ATP produced by enhanced glycolytic flux can partially supply cellular energy demands (*Iyer et al., 1998*; *Seagroves et al., 2001*; *Semenza, 2012*).

A pioneering study using embryonic *Xenopus* retina revealed that neural progenitor cells (NPCs) are less reliant on oxidative phosphorylation for ATP production than are non-dividing differentiated neurons, and the transition from glycolysis to oxidative phosphorylation is tightly coupled to neuronal differentiation, though the exact molecular basis underlying the transition is unknown (*Agathocleous et al., 2012*). Studies in cardiomyocytes provide an example of how a metabolic transition is regulated during development (*Leone and Kelly, 2011*). Around the postnatal stage, cardiomyocytes exit from the cell cycle and gradually enter a maturation process; mitochondrial oxidative activity increases concurrently with elevated expression of mitochondrial genes. The key transcription factors involved are PPARα and its coactivator PGC-1α, which control a broad range of metabolic and mitochondrial genes. PGC-1α may also play a key role in neuronal metabolism, as PGC-1α knockout mice show obvious neurodegenerative pathology (*Lin et al., 2004*).

Neuronal differentiation from human NPCs derived from embryonic stem cells or induced pluripotent stem cells (iPSCs) is able to recapitulate the in vivo developmental process and has been successfully used to model a variety of neurological diseases (*Qiang et al., 2013*). We used this neuronal differentiation model to explore neuronal metabolic differentiation. The disappearance of HK2 and LDHA, together with a PKM2 splicing shift to PKM1, marks the transition from aerobic glycolysis in NPCs to oxidative phosphorylation in neurons. The protein levels of c-MYC and N-MYC,

which are transcriptional activators of HK2, LDHA and PKM splicing, decrease dramatically. Constitutive expression of HK2 and LDHA results in neuronal cell death, indicating that turning off aerobic glycolysis is essential for neuronal differentiation. The metabolic regulators PGC-1α and ERRγ increase significantly upon differentiation; and their up-regulation is required for maintaining the expression of TCA and mitochondrial respiratory complex genes, which, surprisingly, are largely unchanged compared to NPCs, revealing distinct transcriptional regulation of metabolic genes in the proliferation and post-mitotic differentiation states. Mitochondrial mass increases proportionally with neuronal mass growth, indicating an unknown mechanism linking neuronal mitochondrial biogenesis to cell size. In addition, OGDH, a key enzyme in the TCA cycle, has a novel and conserved neuronal splicing shift, resulting in the loss of a calcium binding motif.

## Result

### Transcription profiling of neuronal differentiation from human NPCs

NPCs were derived from iPSCs reprogrammed from the human BJ male fibroblast line. The protocol for NPC establishment and neuronal differentiation is outlined in *Figure 1—figure supplement 1*. To obtain NPC lines of high purity, colonies containing neural rosettes were manually selected and picked as described in Materials and methods and *Figure 1—figure supplement 2*. The identity and purity of NPCs were examined by anti-Sox2 and Nestin staining (*Figure 1A*). Only high-quality NPC lines containing more than 90% Sox2 and Nestin double-positive cells were used for experiments. After 3 weeks of differentiation, a majority (~85%) of cells expressed the neuronal marker MAP2 (*Figure 1B*). Although rare at 3 weeks, glial cells emerged and proliferated after 4–5 weeks; therefore, 3-week neuronal cultures were used to represent a population of developing neurons. Consistent with a previous study (*Johnson et al., 2007*), neurons after ~4 weeks could be induced to fire multiple action potentials (*Figure 1—figure supplement 3*).

To profile the transcriptomes of NPCs and 1- and 3-week-old differentiated neurons, RNA-seq experiments were performed with two NPC lines established from independent BJ iPSC clones. For each time point, two experimental duplicates were used for each NPC line and their differentiated neurons. The sequencing reads were mapped to a human reference genome by STAR and assembled with Cufflinks (*Dobin et al., 2013*; *Trapnell et al., 2012*). The median sequencing read yield for each sample was ~35 million 100-base reads, of which ~85% were mapped. The FPKM (Fragments Per Kilobase of transcript per Million mapped reads) value calculated by the Cufflink algorithm was used to represent the gene expression level (*Trapnell et al., 2010*). The top 200 genes up- and down-regulated during neuronal differentiation were clustered and are presented as a heatmap (*Figure 1C*). Most of the down-regulated genes were cell cycle related, whereas up-regulated genes were enriched in neuronal function pathways, including synapse formation, transmission of nerve impulses, gated channel activity and neuronal projections (*Figure 1D*). Neuron-specific genes such as NEUROD2, MYT1L, MAPT (Tau), SYN1 (synapsin), CHD5 (*Egan et al., 2013*) were significantly increased (*Figure 1E*), whereas the markers associated with proliferating NPCs, LIN28A/B and FOXM1 (*Cimadamore et al., 2013*; *Karsten et al., 2003*), were considerably decreased (*Figure 1F*). These data indicate that our neuronal differentiation model is reliable.

### Gene expression of glycolysis and TCA enzymes during neuronal differentiation

Through the 10 enzymatic steps of glycolysis, glucose is metabolized to pyruvate. Subsequently, glycolytic pyruvate enters into the mitochondrial TCA cycle to generate NADH and FADH2, which are utilized by mitochondrial OXPHOS complexes to generate ATP. For most of the glycolytic reactions, there are multiple paralogous enzymes, but according to our RNA-seq data, for a majority of the reactions only a single enzyme was predominantly expressed in NPCs or neurons (*Figure 1—figure supplement 4*). As shown in *Figure 1G*, expression of the majority of glycolysis genes was decreased in neurons, particularly enzymes acting at the steps after fructose 1,6 phosphate formation, i.e., ALDOA to LDHA/B, which declined to half to one-third of their level in NPCs. An exception was ENO2, a known neuron-specific gene, which increased by ~two fold. GLUT1/3 and HK2, encoding glucose transporters and hexokinase, respectively, dropped ~ten fold. Notably, HK1, the other hexokinase, and PFKM, a phosphofructokinase, functioning at the first two irreversible steps of

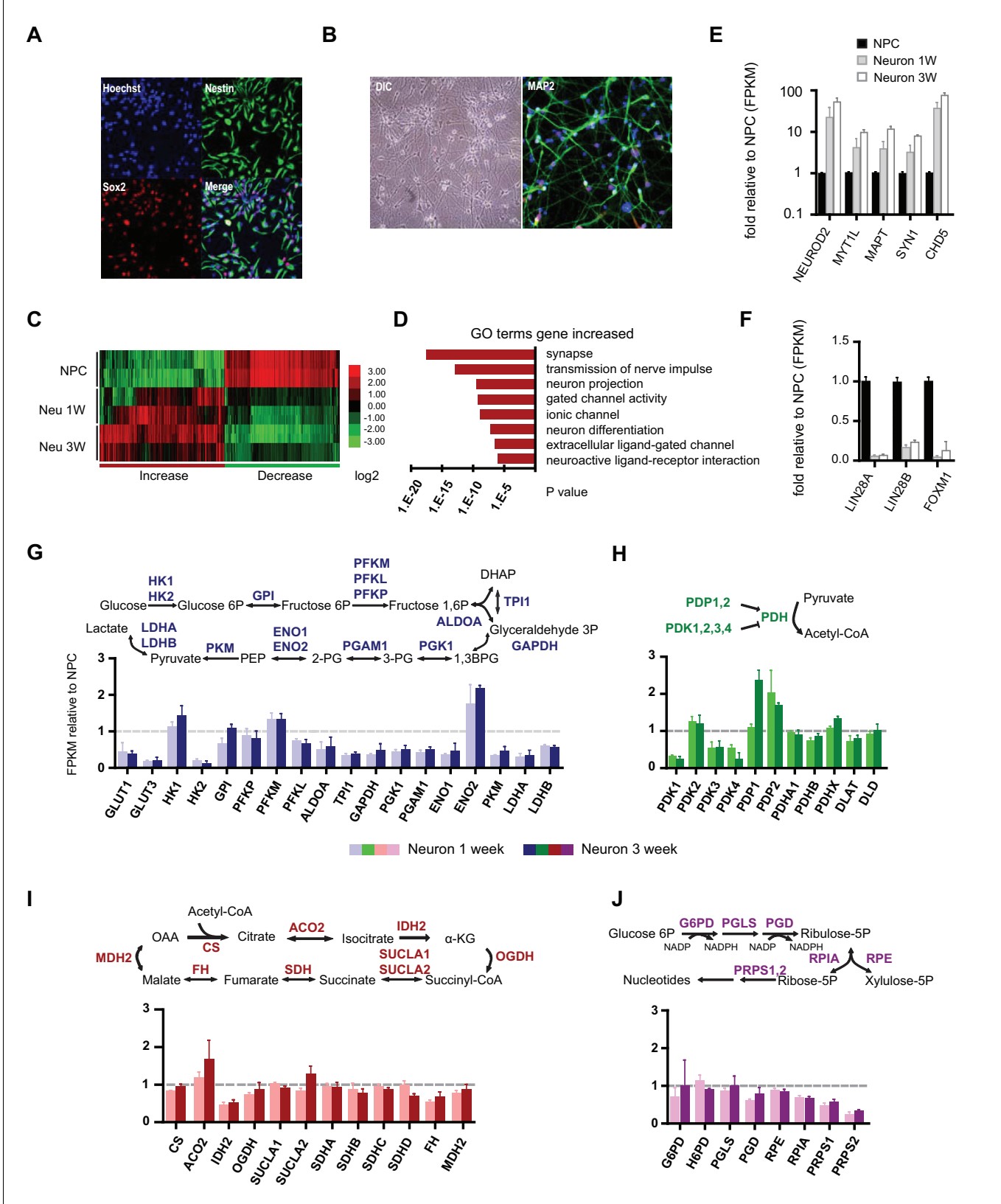

**Figure 1.** Gene expression of glycolysis and TCA during neuronal differentiation. (**A**) Human BJ iPSC-derived NPCs showed homogeneous expression of the NPC markers Nestin and Sox2. (**B**) The left panel shows NPC-derived neurons after 3 weeks of differentiation; right panel shows staining of

*Figure 1 continued*

MAP2, a neuronal marker. (**C**) The top 200 up- and down-regulated genes during neuronal differentiation are depicted by a heatmap. Red and green intensities indicate fold increases and decreases, respectively, in gene expression (expressed as log2). (**D**) GO term analysis of genes up-regulated during neuronal differentiation. The top eight GO term biological process categories obtained are ranked by p-value. (**E**) The FPKM values of known neuron-specific genes are shown as log10-fold change. (**F**) The fold changes of FPKM values of proliferating NPC markers are shown. Bars represent mean ± SD of four RNA-seq replicates for NPCs and neurons differentiated at 1 and 3 weeks. (**G**, **H**, **I** and **J**) Relative expression levels of the key metabolic genes in glycolysis, tricarboxylic acid cycle (TCA), pyruvate dehydrogenase (PDH) complex and pentose phosphate pathway. Bars show the mean of FPKM values of differentiated neurons at 1 and 3 weeks relative to FPKM values of NPCs. Error bars represent SD of four RNA-seq replicates at each time point. Abbreviations, dihydroxyacetone phosphate (DHAP); 1,3-bisphosphoglyceric acid (1,3 BPG); 3-phosphoglyceric acid (3PG); phosphoenolpyruvic acid (PEP), α-ketoglutarate (α-KG); oxaloacetate (OAA). (*Figure 1—sourcer data 1*).

The following source data and figure supplements are available for figure 1:

**Source data 1.** FPKM values of glycolysis, TCA, PDH and pentose phosphate pathway in NPCs and differentiated neurons.

**Figure supplement 1.** The outline of the protocol used to differentiate neurons from iPSCs (upper panel).

**Figure supplement 2.** Colonies containing neural rosettes, and type 4 colony produces NPCs of the best quality.

**Figure supplement 3.** Electrophysiological study of BJ 5-week neurons.

**Figure supplement 4.** FPKM values of paralogous genes in glycolysis.

**Figure supplement 5.** Metabolites quantified by gas chromatography mass spectrometry (GC-MS).

**Figure supplement 5—source data 1.** Metabolites quantified by GC-MS.

**Figure supplement 6.** Oxygen consumption rate (OCR) analysis by Seahorse extracellular flux analysis.

**Figure supplement 6—source data 1.** OCR measurement by Seahorse extracellular flux analysis.

**Figure supplement 7.** The glucose and lactate concentrations in the medium growing NPCs and 3-week neurons were quantified by YSI 2950 metabolite analyzer.

**Figure supplement 7—source data 1.** The ratio of lactate production/glucose consumption.

glycolysis, exhibited minor increases. In contrast to a general decrease in glycolysis genes, TCA genes remained at the same level during differentiation; an exception was isocitrate dehydrogenase, IDH2, the mitochondrial isoform of IDH, which dropped nearly 50% (*Figure 1I*).

The mitochondrial pyruvate dehydrogenase complex (PDC) converts pyruvate to acetyl-CoA for TCA cycle entry. Expression of the genes encoding the subunits of PDC itself, including PDHA1, PDHB, PDHX, DLAT, and DLD, showed no significant changes (*Figure 1H*). The enzymatic activity of PDC is regulated by phosphorylation, being inhibited by a family of protein kinases (PDKs) and activated by a family of protein phosphatases (PDPs). Except for PDK2, RNA levels of the other three PDKs declined significantly during neuronal differentiation; in contrast, PDP1 RNA levels increased ~2.5 fold at 3 weeks of differentiation (*Figure 1H*). The overall changes in PDC kinase and phosphatase levels would favor increased PDC complex activity in neurons, and indeed PDH Ser300 phosphorylation, a major inhibitory site, decreased considerably during neuronal differentiation consistent with an increase in PDC activity (see Figure 3A). The pentose phosphate pathway generates NADPH and ribose-5-phosphate. NADPH is used in reductive biosynthesis reactions; ribose-5-phosphate is a precursor for nucleotide synthesis. Overall, the genes at the beginning steps involved in NADPH production showed unchanged expression levels during differentiation, but the levels of the ones on the branch required for nucleotide synthesis decreased, consistent with neuronal postmitotic status (*Figure 1J*).

## Glycolytic and TCA metabolites and oxygen consumption rate in NPCs and neurons

To further define the metabolic profiles of glycolysis and TCA pathways, we measured representative glycolytic and TCA metabolites by gas chromatography mass spectrometry (GC-MS) (*Figure 1—figure supplement 5*). Consistent with the overall decreased gene expression in glycolysis and glucose transporters, the levels of 3-phosphoglyerate (3PG) and pyruvate in neurons dropped to ~52% and 32% of those in NPCs. The levels of TCA intermediates in neurons were also lower than those in NPCs: citrate was ~32% of that in NPCs, α-ketoglutarate was ~17%, fumarate was ~30%, and malate was ~22%; succinate was relatively higher, at ~62%, which may be due to the decreased protein level of SDHB in complex II, which catalyzes the conversion from succinate to fumarate (see Figure 5C). It should be noted that absolute metabolite levels do not directly reflect true metabolite flux, i.e., the observed reduction in TCA cycle intermediates does not necessarily mean that carbon flux through the TCA cycle is decreased. To directly estimate mitochondrial oxidative activity, the oxygen consumption rate (OCR) was measured in 3-week differentiated neurons and in early passage (P3) NPCs using the Seahorse extracellular flux analyzer (*Figure 1—figure supplement 6*). The basal and maximum OCRs of NPCs were ~82 and 130 (pmoles/min/10 µg), whereas those in 3-week differentiated neurons were ~98 and 183, significantly higher than in NPCs, consistent with increased TCA cycle flux. The lactate production/glucose consumption ratio was measured to estimate the fraction of pyruvate that was converted into lactate and secreted or else transported into mitochondria. As shown in *Figure 1—figure supplement 7*, the ratio was ~1.61 lactate/glucose for NPCs, while in neurons, it was ~0.35, consistent with a significant decrease in aerobic glycolysis in neurons.

## Neuron-specific splicing of PKM and OGDH

Besides expression level, mRNA splicing also influences the activities of several metabolic enzymes. Pyruvate kinase (PKM) is such an example. Alternative RNA splicing generates two isoforms from the *PKM* gene, PKM1 and PKM2, which have their own exclusive exons, exon 9 for PKM1 and exon 10 for PKM2. PKM1 is normally found in tissues where oxidative phosphorylation is preferentially used to generate ATP, whereas PKM2 is expressed in proliferating cells with high anabolic activity (*Imamura and Tanaka, 1972*). PKM1 is constitutively active, while, PKM2 is subjected to allosteric regulation and can adopt a low or high activity state (*Anastasiou et al., 2012*). It is still unclear how PKM1/2 isoforms control cell metabolism in proliferation and non-proliferation states. Replacing PKM2 with PKM1 in cancer cells reduces lactate production and increases oxygen consumption (*Christofk et al., 2008*), indicating that PKM2 may favor high flux glycolysis. A recent study reveals that PKM1 expression does not affect upstream glycolytic intermediates but significantly reduces nucleotide biosynthesis (*Lunt et al., 2015*).

To find out whether changes in alternative mRNA splicing occurred in glycolysis, TCA and mitochondrial respiratory complex genes during neuronal differentiation, the distributions of exonic RNA-seq reads were manually compared between NPCs and neurons using Integrative Genomics Viewer (*Robinson et al., 2011*). Two genes were found to exhibit such mRNA splicing shifts (*Figure 2A*). One was PKM in glycolysis; the other was OGDH (α-ketoglutarate dehydrogenase) in the TCA cycle. NPCs mainly expressed PKM2, whereas neurons expressed PKM1, as confirmed by RT-PCR with isoform-specific primers (*Figure 2B*). In contrast, primary human astrocytes only expressed PKM2. This result was consistent with a recent finding that mouse neurons exclusively expressed PKM1, whereas astrocytes only expressed PKM2 (*Zhang et al., 2014*). Our results established that the splicing shift from PKM2 to PKM1 occurred during neuronal differentiation from NPCs. PKM splicing or the PKM2/PKM1 ratio is known to be controlled by the expression levels of nuclear ribonucleoprotein (hnRNP) proteins, hnRNPI, hnRNPA1 and hnRNPA2, and in cancer cells, these proteins are expressed at high levels and bind repressively to the sequences flanking exon 9, favoring the exon 10 addition thus generating more PKM2 (*David et al., 2010*; *Clower et al., 2010*). Consistent with this model, during neuronal differentiation, hnRNPI, hnRNPA1 and hnRNPA2 RNA levels decreased to nearly 40% compared to NPCs (*Figure 2C*).

OGDH encodes the E1 subunit of the α-ketoglutarate dehydrogenase complex, catalyzing the conversion of α-ketoglutarate to succinyl-CoA in the TCA cycle. Mitochondrial calcium increases can further boost its activity (*Denton, 2009*). OGDH in NPCs (OGDH1) uses exon 4, whereas the isoform in neurons, here called OGDHneu, uses two adjacent exons (*Figure 2A*). The alternative splicing of

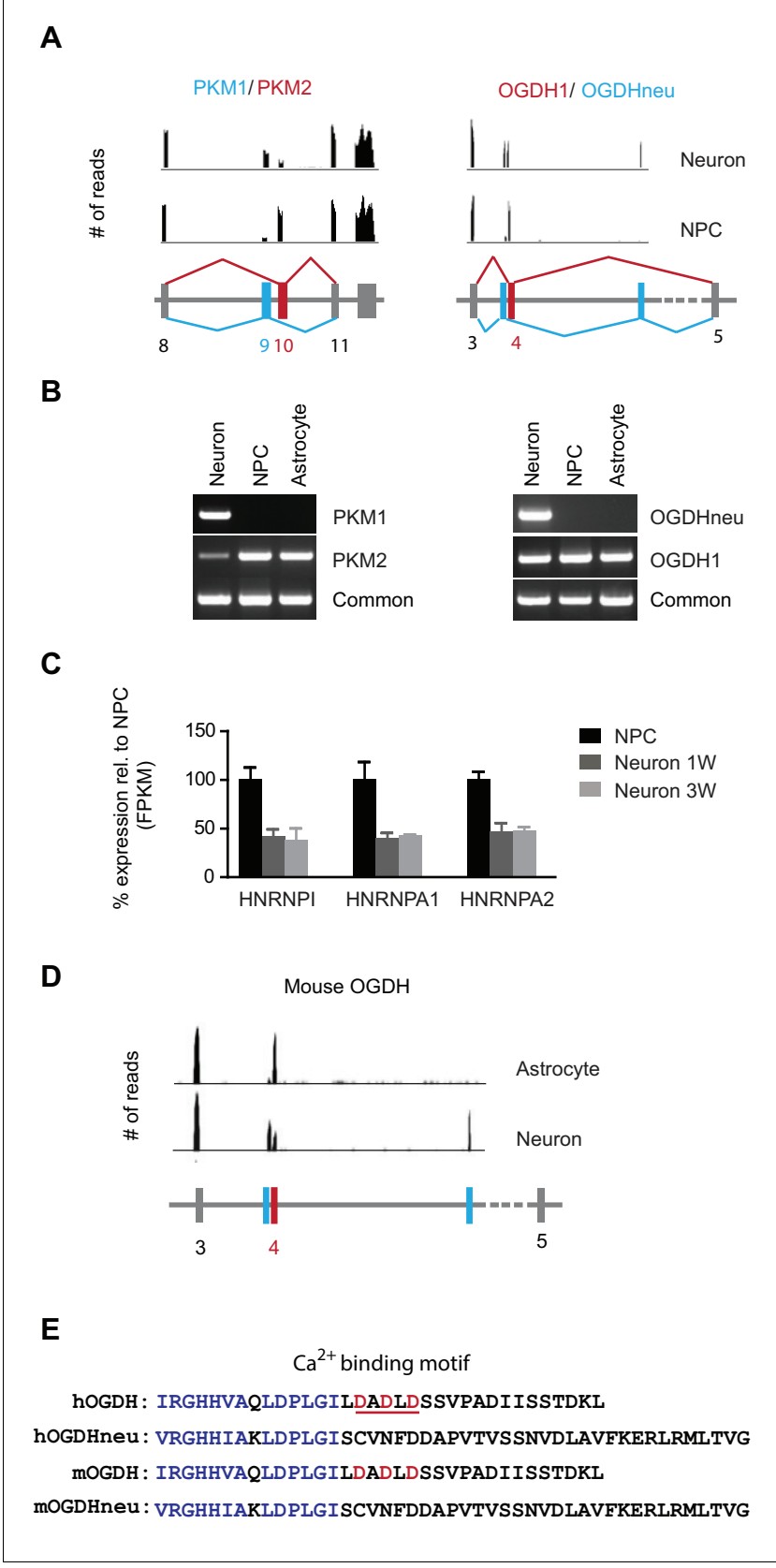

**Figure 2.** Neuron-specific splicing of PKM and OGDH. (**A**) The upper panel shows the RNA-seq reads obtained from NPCs and neurons mapped to the *PKM* and *OGDH* chromosome locus using Integrative Genomics Viewer. *Figure 2 continued on next page*

*Figure 2 continued*

The lower schematic diagram depicts the organization of exons near the cell type-specific splicing site. Red box and line represent exon splicing unique to NPCs; blue ones represent that unique to neurons. PKM2 and OGDH1 are unique to NPCs; PKM1 and OGDHneu are predominantly for neurons. (B) Validation of PKM, OGDH splicing by PCR. Primers were designed to amplify the unique splicing region and common region of PKM1/2 and OGDH1/neu mRNA. PCR was carried out with cDNA prepared from NPCs, neurons at 3 week and primary human astrocytes. (C) The fold changes of FPKM values of hnRNPI, hnRNPA1 and hnRNPA2 are shown. Bars represent mean ± SD of four RNA-seq replicates for NPCs and neurons differentiated at 1 and 3 week. (D) The RNA-seq reads obtained from purified mouse astrocytes and neurons mapped to the *PKM* and *OGDH* chromosome locus using Integrative Genomics Viewer. The original RNA-seq data were from *Zhang et al. (2014)*. (E) Alignment of amino acid sequences encoded by the alternative exons of human and mouse OGDH1/neu. OGDH1 contains a calcium-binding motif underlined in red, absent in OGDHneu. (*Figure 2—source data 1*).
The following source data is available for figure 2:

**Source data 1.** FPKM values of hnRNPI, hnRNPA1 and hnRNPA2.

OGDH was confirmed by RT-PCR with isoform-specific primers (*Figure 2B*). Primary human astrocytes also only expressed OGDH1 (*Figure 2B*). Different from the more exclusive PKM splicing, there was still considerable expression of OGDH1 in neurons. OGDHneu splicing is conserved in mouse neurons, whereas mouse astrocytes, like human astrocytes, only expressed OGDH1 (*Figure 2D*). As a result of this splice switch, a stretch of 34 amino acids in OGDH1, containing a $Ca^{2+}$-binding motif (*Armstrong et al., 2014*), is replaced by 45 amino acids specific to OGDHneu (*Figure 2E*). In a recent study, *Denton et al. (2016)* also detected this splice form of OGDH in the whole brain tissue and pancreatic islets, calling it LS1, and, importantly, they confirmed that LS1 isoform is insensitive to $Ca^{2+}$. Note, that we did not find the OGDHneu isoform in RNA-seq data from cardiomyocytes, indicating that, unlike PKM1, OGDHneu is not specific for cells that preferentially use oxidative phosphorylation. Calcium is used in a variety of neuronal functions. We suspect that OGDH neuron-specific splicing may avoid a prolonged activation of the OGDH complex as a result of frequent intracellular calcium increases.

## HK2 and LDHA were off during neuronal differentiation

Next, we surveyed the protein levels of representative glycolysis and TCA enzymes by immunoblotting (*Figure 3A*). Equal amounts of protein extracts from NPCs and neurons at days 3, 7, and 21 of differentiation (D3, D7, D21, respectively) were resolved by SDS-PAGE. The protein levels of the enzymes examined were largely consistent with their RNA levels. HK2 and LDHA mRNAs were about 15% and 25% of those in NPCs, but strikingly, both proteins almost completely disappeared upon differentiation (*Figure 3A*). To further confirm HK2 and LDHA expression patterns, immunostaining was done on NPCs and 3-week differentiated neurons. Consistent with the immunoblotting data, HK2 and LDHA could readily be detected in NPCs but not in neurons (*Figure 3B*). HK2 catalyzes the ATP-dependent phosphorylation of glucose to yield glucose-6-phosphate, an irreversible step in glycolysis dictating the amount of glucose entry. LDHA catalyzes the conversion pyruvate into lactate at the last step of extended glycolysis, diverting pyruvate from the mitochondrial TCA cycle and recycling $NAD^+$ to sustain high flux glycolysis. The drastic decreases in HK2 and LDHA, the two key enzymes supporting aerobic glycolysis (*DeBerardinis and Thompson, 2012*; *Dang, 2012*), are consistent with neurons not having high flux aerobic glycolysis and exhibiting low lactate production. For the TCA enzymes, the levels of citrate synthase (CS), isocitrate dehydrogenase (IDH2) and succinyl-CoA ligase β subunit (SUCLA2) were constant during differentiation (*Figure 3A*), even though IDH2 showed ~50% decrease at the RNA level. Based on the RNA and protein data, we conclude that there are no significant changes in TCA gene expression in neurons compared to NPCs. To explore if the loss of HK2 and LDHA expression during neuronal differentiation is conserved, we checked HK2 and LDHA expression during differentiation of mouse neuroprogenitor cells derived from embryonic stem cell. As shown in *Figure 3—figure supplement 1*, both HK2 and LDHA mRNA levels dropped, falling to ~8.5% and ~11% of the level in NPCs, respectively, and, consistently, their protein levels were also greatly decreased. In contrast, as in human cells, sustained levels of HK1

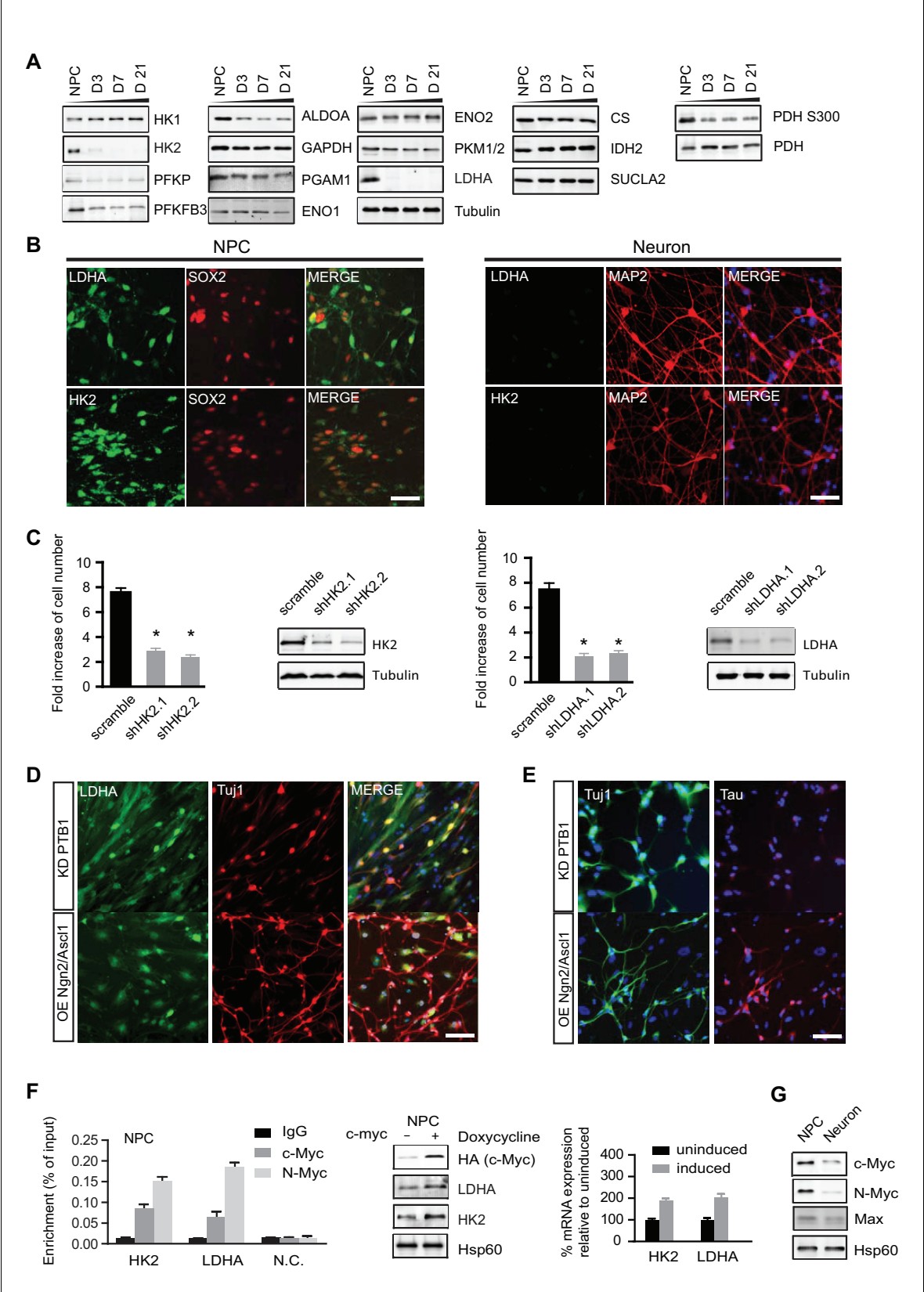

**Figure 3.** Characterization of HK2 and LDHA in NPCs, differentiated neurons and neurons directly converted from fibroblasts. (**A**) Immunoblotting analysis of the representative metabolic genes in glycolysis, tricarboxylic acid cycle (TCA) pathways. 20 μg of protein lysate from NPCs and from

*Figure 3 continued on next page*

*Figure 3 continued*

neurons differentiated for 3, 7 and 21 days (D3, D7, D21) were loaded. (**B**) Immunostaining analysis of HK2 and LDHA in NPCs and 3-week neurons. (**C**) Effects of HK2 or LDHA knockdown on NPC proliferation. NPCs at early passage (P2) were seeded in 24-well plates one day before infection. The NPC number was determined at 5 days after infection with lenti-shRNA virus against HK2 or LDHA. Two effective shRNA lentiviral vectors targeting different regions of HK2 or LDHA were used. Scramble shRNA vector was used as control. Error bars represent ± SD, n= 3. The knockdown efficiency was confirmed by immunoblotting. (**D, E**) Immunostaining analysis of LDHA in neurons directly converted from fibroblasts. Two protocols were applied: one was to knockdown PTB1, a single RNA-binding protein, and the other was to overexpress proneuronal transcription factors Ngn2 and Ascl1. Tuj1 (ß-III tubulin) and Tau were stained as early and mature neuronal markers, respectively. The above experiments were repeated at least three times. (**F**) ChIP analysis of *HK2* and *LDHA* promoters using anti-cMYC or N-MYC antibodies and rabbit IgG as control. Chromatin were prepared from NPCs. The enrichment values are shown as percentage normalized to input. N.C. stands for non-specific control. Bars are mean ± SD, n= 3. Immunoblotting and real time PCR analysis of HK2 and LDHA expression in NPCs with inducible c-MYC. mRNA expression levels of HK2 and LDHA relative to those from non-induction control were calculated after normalization to β-actin. Bars are mean ± SD, n= 3. (**G**) Immunoblotting analysis of c-MYC, N-MYC and Max in NPCs and 3-week neurons. (*Figure 3—source data 1*).

The following source data and figure supplements are available for figure 3:

**Source data 1.** Knockdown effect on NPC proliferation and Myc control of HK2 and LDHA in NPCs.
**Figure supplement 1.** Immunoblotting analysis of mouse HK1, HK2 and LDHA in mouse NPCs derived from embryonic stem cell (ES-E14TG2a), 2-week differentiated neurons and embryonic neurons at E18.
**Figure supplement 1—source data 1.** RT-PCR analysis of HK1, HK2 and LDHA during mouse neuronal differentiation.
**Figure supplement 2.** Immunoblotting and real time PCR analysis of HK2 expression in neurons with inducible c-MYC.
**Figure supplement 2—source data 1.** Activation of HK2 by ectopic c-Myc expression in neuron.

mRNA and protein, were observed during mouse neuronal differentiation. Similar results were observed in primary mouse embryonic neurons at E18. Thus, it appears that the disappearance of LDHA is, mechanistically, a major switch for downregulating aerobic glycolysis as NPCs differentiate, allowing a transition into a neuronal mitochondrial oxidative phosphorylation state.

Because loss of LDHA considerably impairs hematopoietic stem and progenitor cells during hematopoiesis (*Wang et al., 2014*), we examined whether HK2 and LDHA are required for human NPC propagation by performing knockdown experiments. BJ NPC cells at passage 3 were infected with lenti-shRNAs targeting HK2 or LDHA. For each gene, two shRNA constructs against different mRNA region were used; scramble shRNA was used as control. Depletion of HK2 or LDHA significantly slowed the propagation of NPCs by 2.6~3.5 fold (*Figure 3C*). These results indicate that, as found in cancer cell lines (*Fantin et al., 2006*; *Wolf et al., 2011*), aerobic glycolysis is required for NPC proliferation. The inhibitory effect of HK2 depletion on NPC proliferation is consistent with a recent finding that ablation of HK2 diminishes aerobic glycolysis and disrupts cerebellar granule neuronal progenitor development (*Gershon et al., 2013*).

MYC, a key transcription factor associated with cell cycle progression, directly activates HK2 and LDHA transcription, and the promoter regions of *HK2* and *LDHA* contain MYC binding sites (5'-CA<u>C</u>GTG-3'), also known as E box (*Kim et al., 2004*). MYC also promotes a high PKM2/PKM1 ratio by upregulating heterogeneous nuclear ribonucleoproteins (hnRNP) that regulate PKM alternative splicing (*David et al., 2010*). c-MYC and N-MYC redundantly promote neural progenitor cell proliferation in the developing brain, with the brains of double knockout mice exhibiting profoundly impaired growth (*Wey and Knoepfler, 2010*). As shown in *Figure 3F*, chromatin immunoprecipitation with anti-c-MYC and N-MYC antibodies in human NPCs showed that both MYCs could bind to the *HK2* and *LDHA* promoters in the region containing the E boxes as reported (*Kim et al., 2004*). Consistently, we found that overexpression of c-MYC using a doxycycline-inducible lentivirus upregulated HK2 and LDHA expression in NPCs (*Figure 3F*), indicating that MYC controls HK2 and LDHA expression in these cells similarly to other types of cells previously studied. Both c-MYC and N-MYC protein levels decreased significantly, by ~70% and ~85% respectively, once NPCs differentiated into neurons (*Figure 3G*). Dox-induced expression of c-Myc is sufficient to re-activate HK2 expression in neurons (*Figure 3—figure supplement 2*). Therefore, the decreased expression of

HK2 and LDHA during neuronal differentiation was most likely attributable predominantly to the observed MYC decrease occurring as NPCs exited from the cell cycle and entered the differentiation process.

## LDHA is down-regulated in neurons directly converted from fibroblasts

Functional induced neurons (iN) can be generated directly from fibroblasts by forced expression of neural specific transcription factors, a process also called transdifferentiation (*Vierbuchen et al., 2010*). However, it is unknown whether the metabolism of directly converted neurons is the same as that of NPC-differentiated neurons. To test if LDHA was also turned off in neuronal transdifferentiation, two reprogramming protocols were applied. One was to knock down a single RNA-binding protein PTB (*Xue et al., 2013*), and the other was to overexpress the proneuronal transcription factors Ngn2 and Ascl1 (*Ladewig et al., 2012*). Both approaches generated Tuj1-positive neurons after 3 weeks of transduction. The second protocol using Ngn2 and Ascl1 overexpression generated more iNs, and the Tuj1-positive neurons did not have an LDHA immunofluorescence signal (*Figure 3D*). In contrast, with PTB knockdown, a much higher rate (~80%) of colocalization of LDHA and Tuj1 was observed (*Figure 3D*). Furthermore, while Ngn2/Ascl1-derived iNs expressed Tau, a mature neuronal marker, no Tau expression was detected in iNs from PTB knockdown, suggesting that PTB KD iNs were not as mature as Ngn2/Ascl1-derived iNs (*Figure 3E*). These data established that LDHA was also downregulated in neurons directly converted from fibroblasts, indicating a metabolic resetting during transdifferentiation.

## Shutoff of aerobic glycolysis is critical for neuronal differentiation

To determine if shutoff of aerobic glycolysis is critical for neuronal differentiation, we constitutively co-expressed HK2 and LDHA (Flag-tagged) in NPCs using a genome-integrating transposon-based vector, and examined the effects on neuronal differentiation. Their expression was confirmed by immunoblotting (*Figure 4A*), and immunostaining with anti-FLAG antibody revealed that ~80% of NPCs have discernible exogenous LDHA expression (*Figure 4B*). Three-week neuronal cultures differentiated from NPCs constitutively expressing HK2 and LDHA showed a significant fraction of GFAP-positive glial cells, ~40% of total cells, compared to only ~4% GFAP-positive cells in neuronal cultures from control vector-transduced NPCs. Consistently, real-time PCR analysis of GFAP abundance in total mRNA also showed a nearly eight-fold increase (*Figure 4C*). The neuronal cultures from NPCs constitutively expressing HK2 and LDHA had ~40% condensed nuclei, an indicator of apoptotic cells, which was much higher than control, ~6%, and such increased cell death was not observed in NPCs (*Figure 4D*). The LDHA signal was clearly detected in GFAP-positive cells with anti-LDHA antibody (*Figure 4E*) or anti-FLAG antibody staining (*Figure 4H*), but there was no colocalization with MAP2, a neuronal marker. Anti-LDHA and FLAG staining also showed punctate patterns (*Figure 4E and H*), which were associated with punctate MAP2 staining and condensed nuclei (*Figure 4F and G*), indicating that dead cells arose from neurons with exogenous LDHA expression. The results suggest that neurons cannot tolerate constitutive HK2 and LDHA expression and die during differentiation. Neurons detected in the culture differentiated from NPCs constitutively expressing HK2 and LDHA may have been derived from the fraction of NPCs not expressing HK2 and LDHA. Currently, we do not understand why glial cells increase in the population.

As aerobic glycolysis diverts pyruvate from mitochondrial oxidative phosphorylation for energy production by converting it into secreted lactate, a likely reason for the neuronal death is mitochondrial ATP production deficiency. As neuronal cultures differentiated from NPCs constitutively expressing HK2 and LDHA had a large fraction of GFAP-positive glial cells, it is challenging to make a meaningful comparison of lactate secretion or ATP content at the later stages of differentiation. Therefore, we chose to measure AMPK T172 phosphorylation, a cellular energy sensor, during the early stage of neuronal differentiation. Indeed, AMPK pT172 was clearly increased at day 4 of neuronal differentiation compared to control, but the phosphorylation returned to a level comparable to control at day 21, when extensive cell death had already occurred (*Figure 4I*). Day-4 neuronal cultures constitutively expressing HK2 and LDHA showed ~four fold increase of lactate secretion compared to control cultures (*Figure 4J*), indicating increased aerobic glycolysis. These results imply that turning off aerobic glycolysis is critical to maintain the normal neuronal ATP level. If the neuronal death were due to excessive conversion of pyruvate to lactate resulting in reduced pyruvate entry

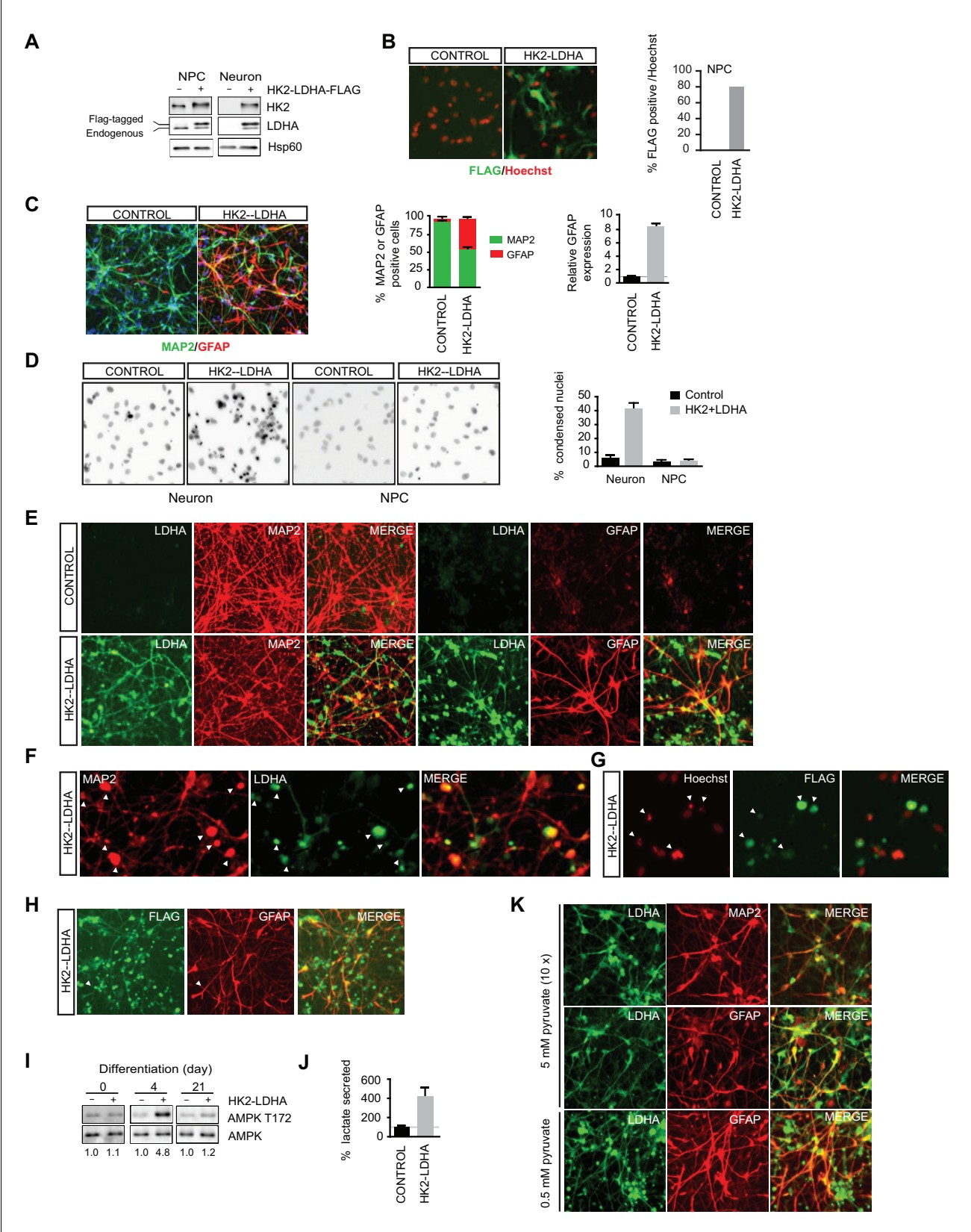

**Figure 4.** Shutoff of aerobic glycolysis is critical for neuronal differentiation. (**A**) Immunoblotting analysis of HK2 and LDHA in NPCs and 3-week neurons constitutively expressing HK2 and LDHA (Flag-tagged). (**B**) Immunostaining of NPCs with anti-FLAG antibody (green), and nuclear staining was done

*Figure 4 continued on next page*

*Figure 4 continued*

with Hoechst (red). The percentage of Flag-positive cells were quantified, and 100 cells were counted for each group. (C) Immunostaining analysis of MAP2 and GFAP in 3-week neurons. The percentage of GFAP and MAP2 cells were quantified, and 100 cells were counted for each group, and three times of neuronal differentiation were included. Bars are mean ± SD, n= 3. The GFAP mRNA abundance in the RNA extracted from neuronal culture was quantified by real-time PCR and normalized to β-actin, and presented as a fold increase compared to neurons differentiated from control NPCs. Bars are mean ± SD, n = 3. (D) Nuclear staining with Hoechst in NPCs and 3-week neurons. The percentages of condensed nuclear were quantified, and 50 cells were counted for each group. Bars are mean ± SD, n= 3. (E) Immunostaining analysis of LDHA, MAP2 and GFAP in 3-week neurons differentiated from NPC constitutively expressing HK2 and LDHA. (F, G) Colocalization of irregular punctated staining of LDHA (green) with MAP2 (red, in F) or condensed nuclear stained with Hoechst (red, in G) in 3-week neurons differentiated from NPC constitutively expressing HK2 and LDHA. (H) Immunostaining analysis of anti-FLAG and GFAP in 3-week neurons differentiated from NPC constitutively expressing HK2 and LDHA. (I) Immunoblotting analysis of AMPK T172 phosphorylation in the cell lysate extracted from day 4 and day 21 neuronal culture differentiated from NPCs expressing HK2 and LDHA. The AMPK T172 phosphorylation was quantified after normalized to non-phosphoAMPK, and presented as fold increase. (J) Lactate in medium from day-4 neuronal culture differentiated from NPC expressing HK2 and LDHA were quantified and normalized by protein content, and presented as percentage compared to those from control neurons, bars are mean ± SD, n= 3. (K) Immunostaining analysis of LDHA, MAP2 and GFAP in 3-week neurons differentiated from NPC constitutively expressing HK2 and LDHA. The neuronal differentiation medium used contained 5 mM sodium pyruvate, ten fold of the standard concentration (0.5 mM). The above experiments were repeated at least three times. (*Figure 4—source data 1*).

The following source data is available for figure 4:

**Source data 1.** Constitutive expression of HK2 and LDHA is detrimental for neuronal differentiation.

into mitochondria, increasing the level of pyruvate in neuronal differentiation medium might be able to rescue the neuronal death. This was indeed the case; as shown in *Figure 4K*, coexpression of LDHA and MAP2 was readily detected in cells grown in differentiation medium containing 5 mM sodium pyruvate, ten-fold higher than the standard concentration (0.5 mM), and there was no irregular punctate staining of LDHA as seen with standard neuronal differentiation medium (*Figure 4E*). However, GFAP-positive glial cells were still abundant (*Figure 4K*).

## Increased neuronal PGC-1α/ERRγ maintained the transcription of glycolysis, TCA and mitochondrial genes

A transcriptional circuit, including PGC-1 family coactivators, estrogen-related receptors (ERRs) and nuclear respiratory factor (NRF), is responsible for the transcription of metabolic and mitochondrial genes in multiple tissues, such as heart and skeletal muscle (*Scarpulla et al., 2012*). PGC-1α is induced in the neonatal mouse heart concurrent with the increase of mitochondrial energy-producing capacity (*Lehman et al., 2000*). However, the control of mitochondrial biogenesis during neuronal differentiation is still not well understood.

To examine if there were any changes in mitochondrial mass during neuronal differentiation, we first measured mtDNA copy number, a commonly used surrogate of mitochondrial mass, calculated by normalizing to nuclear genomes. As shown in *Figure 5A*, after the first week, the neuronal mtDNA copy number doubled compared to NPCs, and at 3 weeks was increased to ~four fold. To roughly estimate the extent of cell growth, we measured the protein mass of NPCs and 3-week differentiated neurons. Consistent with neurons being larger cells than NPCs, we found that 3-week differentiated neurons contained ~four-fold higher protein mass than NPCs (*Figure 5B*), similar to the increase in mtDNA copy number. When the amounts of mitochondrial respiratory complexes were examined by immunoblotting with equal protein loading of cell lysates, no significant difference was found between NPCs and neurons at 1 and 3 weeks, except for SDHB of complex II, which was consistently ~50% lower in neurons (*Figure 5C*). The same was true for other mitochondrial markers, such as TFAM, Hsp60 and ATP5O (*Figure 5D*). It appears that although mitochondrial mass increases on a per cell basis, mitochondrial density remains largely unchanged during neuronal differentiation.

We went on to analyze the expression profiles of transcription factors involved in mitochondrial biogenesis: PGC-1α levels were significantly increased, ~three fold increase at 1 week and ~four fold at 3 week; ERRγ levels were also upregulated, ~2.8 fold at 1 week and ~3.4 at 3 week, and similar changes were observed in their protein levels (*Figure 5D*). Surprisingly, despite the significant increases in PGC-1α and ERRγ, the expression of the majority of the genes encoding mitochondrial

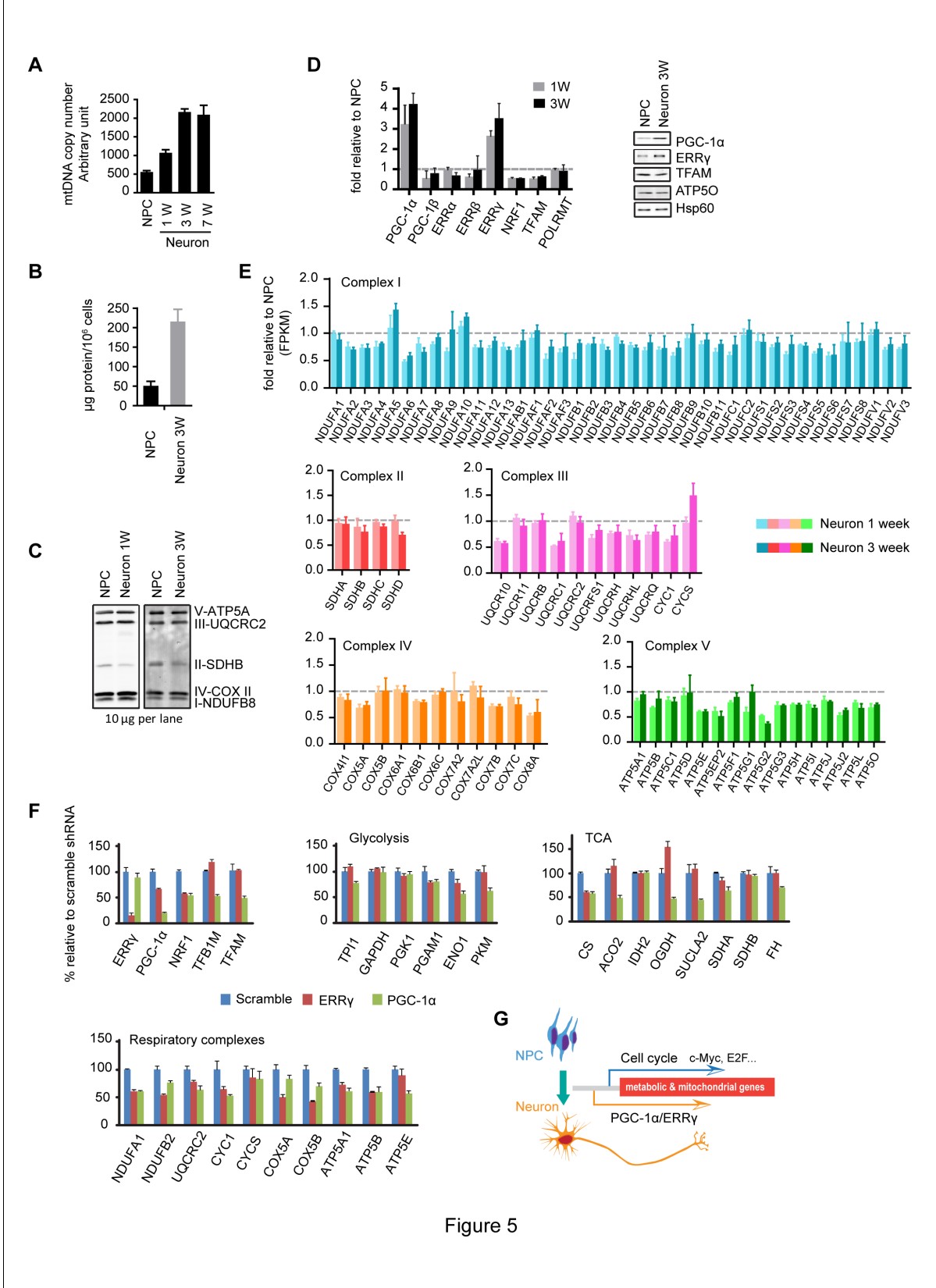

Figure 5

**Figure 5.** Increased neuronal PGC-1α /ERRγ maintain the transcription of metabolic and mitochondrial genes during neuronal differentiation. (**A**) mtDNA copy number was measured during neuronal differentiation. Real-time PCR was done with NPCs and neurons differentiated at 1, 3, and 7

*Figure 5 continued on next page*

*Figure 5 continued*

weeks. Bars represent mean ± SD n= 3. (**B**) Measurement of protein mass content of NPCs and neurons at 3 weeks normalized as in per million cells. Bars represent mean ± SD, n =3. (**C**) Immunoblotting analysis of the representative component of each mitochondrial respiratory complex. 10 µg protein lysate from NPCs and neurons at 1 and 3 weeks were loaded in SDS-PAGE gel. (**D**) Expression changes of main transcription factors involved in the transcription of metabolic and mitochondrial genes. Bars show the mean of FPKM values of differentiated neurons at 1 and 3 weeks relative to those of NPCs. Error bars represent SD of four RNA-seq replicates at each time point. Immunoblotting analysis confirmed the upregulation of PGC-1α and ERRγ. (**E**) Relative expression levels of genes encoding the mitochondrial respiratory complexes. Bars show the mean of FPKM values of differentiated neuron at 1 and 3 weeks relative to those of NPCs. Error bars represent SD of four RNA-seq replicates at each time point. (**F**) Effects of PGC-1α or ERRγ knockdown on the gene expression of glycolysis, tricarboxylic acid cycle (TCA) and mitochondrial respiratory complexes. Neurons differentiated at 3 week were infected with lenti-shRNA virus against PGC-1α or ERRγ, and the total RNA was extracted 5 days after infection. Two effective shRNA lentiviral vectors targeting different regions of PGC-1α or ERRγ were used. Scramble shRNA vector was used as control. In real-time PCR experiments, the relative mRNA expression levels in neurons depleted of PGC-1α or ERRγ to scramble control were calculated after normalization to β-actin. Bars are mean ± SD, n= 3. Similar results were obtained for both shRNA knockdown constructs. (**G**) A hypothetical model depicting distinct transcriptional regulation of metabolic genes in the proliferation and post-mitotic differentiation states. (*Figure 5—source data 1*).

The following source data and figure supplements are available for figure 5:

**Source data 1.** PGC-1α and ERRγ maintain the metabolic gene expression during neuronal differentiation.
**Figure supplement 1.** The fold changes of FPKM values of UCP2 are shown.
**Figure supplement 1—source data 1.** UCP2 expression during neuronal differentiation.

respiratory complexes was either slightly decreased or unchanged in neurons (*Figure 5E*). At first glance, this was confusing, because overexpression of PGC-1α in cardiac cells triggers the upregulation of hundreds of genes in glycolysis, TCA and mitochondrial respiratory complexes (*Rowe et al., 2010*). To examine whether the increased PGC-1α and ERRγ affected TCA and mitochondrial OXPHOS gene expression, knockdown experiments were performed. Three-week neurons were infected with lenti-shRNAs targeting PGC-1α or ERRγ. For each gene, two effective shRNA constructs against different mRNA regions were used, and scramble shRNA was used as control. We found that PGC-1α was required to maintain the levels of ENO1 and PKM in glycolysis; CS, ACO2, OGDH and SUCLA2 in TCA cycle; TFAM and TFBM1 in mtDNA replication and transcription, and NRF1 (nuclear respiratory factor-1), a key transcription factor that activates nuclear genes encoding respiratory complex subunits (*Figure 5F*). PGC-1α not only stimulates the induction of NRF1, but also binds to and coactivates the transcriptional function of NRF1 in muscle cells (*Wu et al., 1999*). Consistently, almost all the representative genes for each mitochondrial respiratory complex exhibited decreased expression in PGC-1α knockdown neurons (*Figure 5F*). In contrast to the broad effect of PGC-1α, ERRγ was mainly required for maintaining expression of NRF1 and mitochondrial respiratory complexes (*Figure 5F*), which is consistent with a recent finding that in neurons the genomic binding sites for ERRγ have extensive overlap with NRFs (*Pei et al., 2015*). Therefore, the increased expression of PGC-1α/ERRγ during neuronal differentiation appears to maintain, rather than increase as assumed, the transcription of metabolic and mitochondrial genes as in NPCs. This observation also implies that there is distinct transcriptional control of metabolic and mitochondrial OXPHOS genes in proliferating versus post-mitotic differentiated cells as discussed below.

## Discussion

Neurons rely on mitochondrial oxidative phosphorylation to meet energy demand. It is unclear how neurons acquire this reliance, particularly from a developmental perspective. Neuronal differentiation from human NPCs is an ideal model to explore this question. In this study, we have uncovered several key molecular events underlying the transition from aerobic glycolysis in NPCs to neuronal oxidative phosphorylation.

# Molecular changes that define the neuronal energy preference for oxidative phosphorylation

A series of transcriptional and protein level changes during neuronal differentiation, as summarized in *Figure 6*, define the neuronal energy preference for oxidative phosphorylation. First, downregulation of LDHA is a key switch for turning off aerobic glycolysis. Lactate dehydrogenase, catalyzing the conversion between pyruvate and lactate, is a tetramer, and the *LDHA* and *LDHB* genes encode the two common subunits A and B respectively, which can assemble into five isozymes - A4, A3B1, A2B2, A1B3, and B4. The A4 isozyme kinetically favors the conversion from pyruvate to lactate, while the B4 isozyme prefer converting lactate to pyruvate (*Cahn et al., 1962*). The critical role of LDHA in aerobic glycolysis has been proven in cancer cells, and LDHA depletion greatly decreases aerobic glycolysis and increases mitochondrial OXPHOS activity (*Fantin et al., 2006*). The absence of LDHA in neurons is consistent with previous immunohistochemistry studies in human brain showing that neurons are exclusively stained with anti-LDHB antibody while astrocytes are stained by both anti-LDHA and LDHB antibodies (*Bittar et al., 1996*). Second, the shift of pyruvate kinase from PKM2 to PKM1 by alternative mRNA splicing is likely to be important. A current hypothesis is that PKM2 expressed in proliferating cells is in a less-active state, thus resulting in accumulation of upstream metabolites that can be used for biosynthetic pathways (*Christofk et al., 2008*; *Eigenbrodt and Glossmann, 1980*), although this model is challenged by a recent finding that PKM1 expression does not decrease upstream glycolytic intermediates but significantly reduces nucleotide biosynthesis (*Lunt et al., 2015*). Third, the decreased expression of HK2, and the GLUT1 and GLUT3 glucose transporters reduce glucose entry, terminating high flux glycolysis. HK1 is still expressed in neurons,

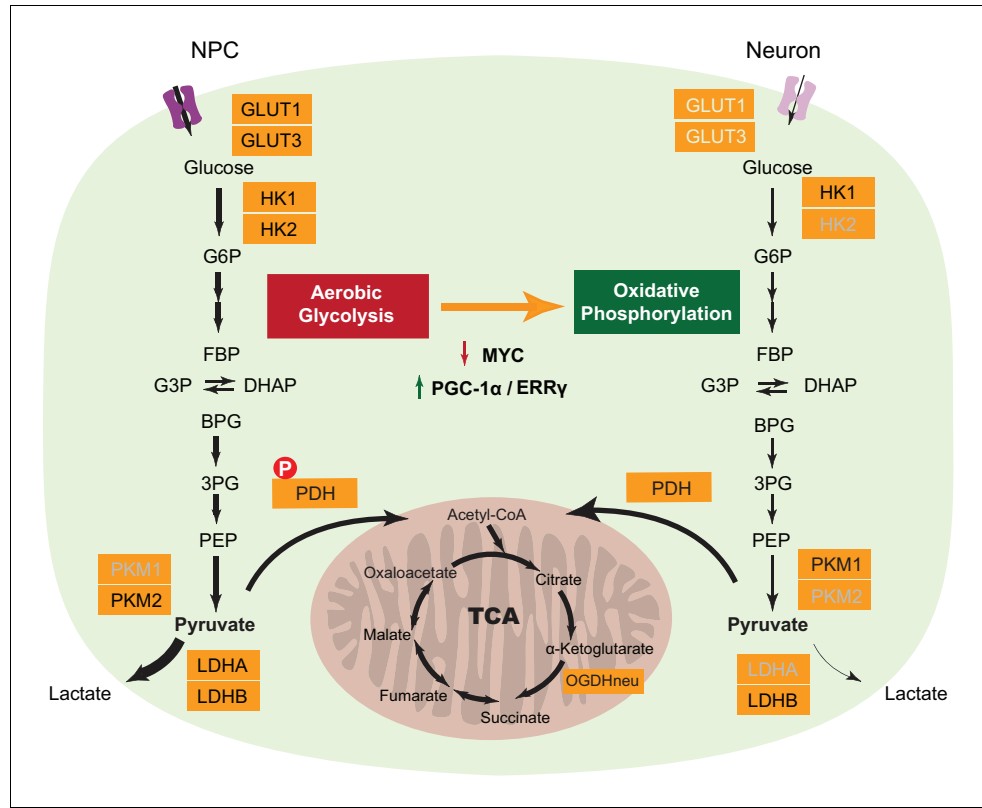

**Figure 6.** A model depicting transcriptional changes of metabolic genes underlying the switch from aerobic glycolysis in NPCs to oxidative phosphorylation in neurons. The genes with decreased expression are dimmed. The width of the arrows indicates increased and decreased pyruvate and lactate utilization at different steps in NPCs and neurons. Abbreviations: glucose 6-phosphate (G6P); fructose 1,6-bisphosphate (FBP); glycerol 3-phosphate (G3P); dihydroxyacetone phosphate (DHAP); 1,3-bisphosphoglyceric acid (BPG); 3-phosphoglyceric acid (3PG); phosphoenolpyruvic acid (PEP).

and this will be needed to provide a low level flux though the glycolytic pathway in order to generate enough pyruvate to feed the TCA cycle. Fourth, the decrease in expression of PDKs and the increase in PDPs may result in more active pyruvate dehydrogenase complexes (PDC), as demonstrated by decreased phosphorylation at Ser 300 of PDH E1 alpha protein (PDHA1), promoting pyruvate entry into the TCA cycle. Consistent with decreased levels of glycolytic genes, 3-phosphoglyceric acid (3PG) and pyruvate levels decline in neurons compared to NPCs. Surprisingly, even though neurons rely on oxidative phosphorylation to generate energy; the expression of TCA and mitochondrial genes is not increased. Thus, it appears that a need to avoid aerobic glycolysis is the major reason underlying neuronal reliance on oxidative phosphorylation.

In addition to the conventional regulators of glycolysis, TCA and oxidative phosphorylation, the critical role of UCP2 (uncoupling protein 2) in promoting aerobic glycolysis and inhibiting oxidative phosphorylation has been established from multiple experimental models (*Pecqueur et al., 2008*; *Samudio et al., 2008*; *Bouillaud, 2009*; *Zhang et al., 2011*). Distinct from UCP1, which uncouples ATP synthesis from the proton gradient by transporting protons into the mitochondrial matrix, UCP2 exports malate and oxaloacetate from mitochondria into the cytosol thus limiting the entry of pyruvate into the TCA cycle (*Vozza et al., 2014*). As demonstrated in human embryonic stem cells and hematopoietic stem cells, UCP2 is critical for maintaining stem cell glycolytic metabolism, and its level decreases during differentiation (*Zhang et al., 2011*; *Yu et al., 2013*). We found that during neuronal differentiation, UCP2 expression levels dropped significantly, and in 3-week neurons the UCP2 level was only 20% of that in NPCs (*Figure 5—figure supplement 1*), indicating that similar mitochondrial metabolism remodeling occurs in neurons.

It should be noted that the expression levels of metabolic enzymes alone do not completely reflect their biochemical/physiological activities, which could be regulated at multiple levels, such as post-translational modifications, which have not been explored in the current study. Moreover, mitochondrial structure (cristae organization and size), dynamics (fusion and fission) and calcium concentration in the mitochondrial matrix are all involved in mitochondrial energy metabolism (*Mishra and Chan, 2014*). Possible changes in these processes need to be investigated to further understand how mitochondrial metabolism is reprogrammed during neuronal differentiation. In addition to transcriptional regulation, protein degradation could also be used to downregulate key metabolic enzymes. For instance, Pfkfb3, the enzyme generating fructose-2,6-bisphosphate, a potent activator of phosphofructokinase, is constantly degraded by proteasome through anaphase-promoting complex/cyclosome (APC/C)-Cdh1 to suppress neuronal glycolysis (*Herrero-Mendez, et al., 2009*). We suspect there might be additional mechanisms of this sort at the protein level accounting for the extremely low levels of HK2 and LDHA protein in neurons.

Even under conditions of energy shortage due to mitochondrial deficiency, neurons cannot turn on aerobic glycolysis genes, such as HK2 and LDHA. In an iPSC-based disease model of maternally inherited Leign syndrome (MILS), an early childhood neurodegenerative disease due to ATP synthase defect, we found that MILS neurons but not their NPCs show severe energy deficiency; MILS NPCs generate more lactate than healthy control NPCs. MILS neurons also have very low expression of HK2 and LDHA, and do not exhibit a significant increase in mitochondrial density (*Zheng et al., 2016*). It appears that neurons cannot use aerobic glycolysis or mitochondrial biogenesis to compensate for energy shortage.

## Shut-off of aerobic glycolysis is critical for neuronal differentiation

To explore the importance of turning off aerobic glycolysis, we attempted to reactivate aerobic glycolysis by constitutive expression of HK2 and LDHA during neuronal differentiation. We found that neurons cannot survive the sustained high levels of HK2 and LDHA that were tolerated by NPCs. Glucose transporter levels dramatically decrease in neurons, which limits glucose uptake and the production of glycolytic pyruvate. Reactivating the conversion of pyruvate to lactate by expression of LDHA would decrease the amount of pyruvate available for mitochondrial oxidation. Indeed, increasing pyruvate in the neuronal differentiation medium prevented the neuronal death observed upon expressing HK2/LDHA, indicating that glycolytic pyruvate deficiency is a major cause of death. Therefore, turning off aerobic glycolysis allows the efficient use of pyruvate for energy production. Many lines of evidence support the conclusion that lactate secreted by glial cells is a critical energy source for neurons in vivo (*Pellerin and Magistretti, 2012*), and blocking neuronal oxidative utilization of lactate affects neuronal survival and even memory formation (*Suzuki et al., 2011*; *Lee et al.,*

*2012*). Obviously, downregulation of LDHA, an enzyme catalyzing the conversion of pyruvate to lactate, would favor the reverse reaction, which is catalyzed by a tetramer composed of LDHB to generate pyruvate from exogenous lactate.

To our surprise, a significant fraction of GFAP-positive glial cells was detected at early times in neuronal cultures differentiated from NPCs constitutively expressing HK2 and LDHA. This phenotype could not be reversed by extra pyruvate in the medium. Interestingly, it has been reported that exposure of NPCs to hypoxia, which boosts aerobic glycolysis, leads to more glial cells during neuronal differentiation (*Xie et al., 2014*). We confirmed this observation in wild-type NPCs (data not shown). Aerobic glycolysis is tightly associated with cellular redox status (*Ochocki and Simon, 2013*). Interestingly, Sirt1, a histone deacetylase and sensor of NADH/NAD$^+$, has been shown to direct the differentiation into the astroglial lineage at the expense of the neuronal lineage (*Prozorovski et al., 2008*). We surmise that enhanced glycolysis may generate a cellular redox status that shifts the lineage choice toward glial cells during differentiation of NPCs.

## Metabolic and mitochondrial gene transcription control in proliferating and post-mitotic differentiated cells

Initially, we were surprised to find that, despite increased neuronal PGC-1α and ERRγ expression, the majority of the genes encoding mitochondrial respiratory complexes and TCA enzymes were not increased. Our knockdown studies revealed that depleting PGC-1α in neurons led to a significant decrease in a wide spectrum of genes in glycolysis, TCA pathway and mitochondrial respiratory complexes. Thus, the increased expression of PGC-1α and ERRγ in neurons is essential to maintain the expression level of these metabolic genes. Although NPCs do not rely exclusively on oxidative phosphorylation to generate energy, mitochondria are used for generation of biosynthetic precursors in these proliferating cells, and mitochondria themselves have to be duplicated for daughter cells. Cell cycle transcription factors, such as MYC and E2F, have been shown to promote metabolic and mitochondrial gene expression. c-MYC-null fibroblasts have diminished mitochondrial mass (*Li et al., 2005*). c-MYC activation in the myocardium of adult mice induces mitochondrial biogenesis and glycolysis but reduces PGC-1α level (*Ahuja et al., 2010*). E2F also upregulates mitochondrial genes, a function conserved from flies to mammals (*Ambrus et al., 2013*). It appears that, during neuronal differentiation, the transcriptional control of mitochondrial genes and metabolic genes shifts from a cell-cycle mode to a post-mitotic neuronal program: in proliferating NPCs, MYC and E2Fs activate the transcription of metabolic genes, while in differentiated neurons, PGC-1α and ERRγ are responsible (*Figure 5G*). This hypothetical model does not exclude the possibility that low levels of PGC-1α and ERRγ take part in metabolic and mitochondrial gene transcription in NPCs together with cell cycle related transcription factors.

During neuronal differentiation, mitochondrial mass increases proportionally with neuronal mass growth, indicating an unknown mechanism linking mitochondrial biogenesis to cell size. Interestingly in this regard, in a high-throughput screen to identify small molecules interfering with mitochondrial abundance, hundreds of compounds were found to be capable of changing the cellular mitochondrial content; the majority of them also change cell size accordingly (*Kitami et al., 2012*), indicating a fundamental relationship between cell size and mitochondrial number. Our finding illustrates an example of this relationship in a normal developmental context.

## Materials and methods

### Immunohistochemistry

Cells were fixed in cold 4% paraformaldehyde in PBS for 10 min. NPCs and neurons were permeabilized at room temperature for 15 min in 0.2% TritonX-100 in PBS. Samples were blocked in 5% BSA with 0.1% Tween 20 for 30 min at room temperature. The following primary antibodies and dilutions were used: goat anti-Sox2 (Santa Cruz Biotechnology, Dallas, TX), 1:200; mouse anti-Nestin (EMD Millipore, Temecula, California), 1:200; rabbit anti-βIII-tubulin (Covance, San Diego, CA), 1:200; mouse anti-βIII-tubulin (Covance), 1:200; rabbit anti-GFAP (Dako, Carpinteria, CA) 1:200; mouse anti-MAP2AB (Sigma-Aldrich, St. Louis, MO), 1:200; rabbit anti-LDHA (Cell signaling, Danvers, MA), 1:200, and rabbit anti-HK2 (Cell signaling), 1:200. Secondary antibodies were Alexa donkey 488 and

568 anti-mouse, rabbit and goat (Invitrogen, Carlsbad, CA), used at 1:1000. Nuclear staining was done with Hoechst (Invitrogen).

## Cell lysate preparation and immunoblotting

Cell lysates were prepared with lysis buffer containing 20 mM Tris (pH 7.5), 150 mM NaCl, 1 mM EDTA, 1 mM EGTA, 1% Triton X-100, 2.5 mM sodium pyrophosphate, 1 mM β-glycerophosphate, 1 mM $Na_3VO_4$, 1 µg/ml leupeptin. 1 mM PMSF was added immediately prior to use. The protein concentration was measured by DC protein assay (Bio-Rad, Irvine, CA). Quick neuron nuclear extract preparation: the cells were rinsed with PBS once, and plates placed on ice and 1 ml of ice cold Buffer A (25 mM Hepes pH 7.0, 25 mM KCl, 0.05 mM EDTA, 5 mM $MgCl_2$, 10% glycerol, 0.1% NP-40, 1 mM DTT) added. The plate was scraped and cells were transferred into an Eppendorf tube, which was centrifuged and rinsed once with Buffer A (no NP-40). The pellet was resuspended in PBS, and 2x SDS PAGE sample buffer added, prior to boiling for 10 min at 95°C. The following primary antibodies and dilutions were used: antibodies against the major glycolysis enzymes were from Cell Signaling sold as glycolysis antibody sampler kits (#8337&12866); all were used at 1:1000; Rabbit anti-TFAM (Cell Signaling) used at 1:1000; Rabbit anti-CS, IDH2, PGC-1α and ATP5O (Abcam, Cambridge, United Kingdom) used at 1:1000, and mouse anti-Sucla2 and goat anti-Hsp60 (Santa Cruz Biotechnology) used at 1:1000. Immunoblotting results were analyzed by Odyssey Imager (Licor, Lincoln, NE) scanning.

## Establishment of NPCs and neuronal differentiation

The establishment of neural progenitor cells from iPSCs and neuron differentiation were performed as previously described (*Brennand et al., 2011*). Human embryonic stem cell (hESC) and iPSC lines were mainly maintained on Matrigel using mTeSR1. For embryoid body formation, hESC and iPSC lines were cultured on a mitotically inactive mouse embryonic fibroblast feeder layer in hESC medium, DMEM/F12 supplemented with 20% knockout serum replacement, 1 mM L-glutamine, 0.1 mM non-essential amino acids, β-mercaptoethanol and 10 ng /ml FGF2. Neural differentiation was induced as follows: hESCs grown on inactivated mEFs were fed N2/B27 medium without retinoic acid for 2 days, and then colonies were lifted with collagenase treatment for 1 hr at 37°C. The cell clumps were then transferred to ultra-low attachment plates. After growth in suspension for 1 week in N2/B27 medium, aggregates formed embryoid bodies, which were then transferred onto polyornithine (PORN)/laminin-coated plates and developed into neural rosettes in N2/B27 medium. After another week, colonies, showing mature neural rosettes with NPCs migrating out from the colony border, were picked under a dissecting microscope, digested with accutase for 10 min at 37°C and then cultured on PORN/laminin-coated plates in N2/B27 medium supplemented with FGF2. This step is critical for the purity of NPCs; only colonies (type 4) showing sufficient maturity as described in *Figure 1—figure supplement 2* were picked. For neuronal differentiation, NPCs were dissociated with accutase and plated in neural differentiation media, 500 ml DMEM/F12 GlutaMAX, 1x N2, 1X B27+RA, 20 ng/ml BDNF (Peprotech, Rocky Hill, NJ), 20 ng/ml GDNF (Peprotech), 200 nM ascorbic acid (Sigma), 1 mM dibutyrl-cyclicAMP (Sigma) onto PORN/laminin-coated plates. For one well of a 6-well plate, 200,000 cells/well were seeded; for one well of a 12-well plate, 80,000 cells were seeded. Neurons can be maintained for 3 months in a 5% CO2 37°C incubator.

## Direct conversion of human fibroblasts into iNs using Ngn2/Ascl1- and KD PTB1-based protocols

For Ngn2/Ascl1-based conversion, postnatal human fibroblasts were transduced with lentiviral particles for EtO and XTP-Ngn2:2A:Ascl1 and selected with G418 (200 µg/ml; Life Technologies) and puromycin (1 µg/ml; Sigma Aldrich). For knockdown PTB1-based conversion, cells were transduced with pLVTHM carrying a shRNA against PTB1 (GCGTGAAGATCCTGTTCAATACTCGAGTATTGAA-CAGGATCTTCACGC) and selected with puromycin. To initiate conversion, Ngn2/Ascl1 or shPTB1 fibroblasts were pooled at high densities and after 24 hr the medium was changed to neuron conversion medium based on DMEM:F12/Neurobasal (1:1) for 3 weeks. The iN conversion medium contains the following supplements: N2 supplement, B27 supplement (both 1x; Gibco), doxycycline (2 µg/ml, Sigma Aldrich), laminin (1µg/ml, Life Technologies, Carlsbad, CA), dibutyryl cyclic-AMP (500 µg/ml, Sigma Aldrich), human recombinant Noggin (150 ng/ml; Preprotech), LDN-193189 (5 µM; Cayman

Chemical Co, Ann Arbor, MI) and A83-1 (5 µM; Stemgent, Cambridge, MA), CHIR99021 (3 µM, LC Laboratories, Woburn, MA), Forskolin (5 µM, LC Laboratories) and SB-431542 (10 µM; Cayman Chemicals Co.). Medium was changed every third day up to 3 weeks. The protocol was adapted from the previous method (*Ladewig et al., 2012*).

## qRT-PCR, mtDNA copy quantification and Chromatin immunoprecipitation assay

Total RNA was isolated using RNeasy kit (QIAGEN, Hilden, Germany). 500 ng of total RNA from each sample was used for cDNA synthesis by MMLV reverse transcriptase; and quantitative real-time polymerase chain reaction (PCR) was performed with SYBR Green Master Mix on ABI 7000 cycler (Applied Biosystems, Foster City, CA) and normalization to β-actin. Primer sequences were listed in *Supplementary file 1*.

mtDNA copy number was estimated by comparing SYBR Green real time PCR amplification of a mitochondrial DNA amplicon, tRNA Leu (UUR), with a nuclear DNA amplicon (β2-microglobulin) from DNA isolated using a Qiagen genomic DNA kit according to the protocol (*Venegas et al., 2011*). The primer pair for mtDNA tRNA Leu: 'CACCCAAGAACAGGGTTTGT' and TGGCCATGGG TATGTTGTTA'; for β2-microglobulin: 'TGCTGTCTCCATGTTTGATGTATCT' and TCTCTGC TCCCCACCTCTAAGT'.

The ChIP assays were performed using ChIP-IT high sensitivity kit (Active Motif, Carlsbad, CA). The polyclonal antibodies against c-MYC, N-MYC, and anti-HA used were purchased from Santa Cruz Biotechnology, Active Motif and Abcam. The real-time PCR primers were designed according to the work by *Kim et al. (2004)* and *(2007)*.

## RNA sequencing

NPCs (at passage 3) and 1- and 3-week differentiated neurons were harvested; total RNA was isolated using RNeasy kit with in-column DNase digestion (QIAGEN). RNA-seq experiments were performed with two NPC lines established from independent iPSC clones. For each time point, two experimental duplicates were used for each independent NPC line and its differentiated neurons. The quality of RNA was quantified by RNA integrity number (RIN) by Bioanalyzer (Agilent, La Jolla, CA); only samples with RIN greater than 8.5 were used for library preparation. Stranded mRNA-seq libraries were prepared from poly(A) RNA after oligo(dT) selection. RNA-seq reactions were done on an Illumina HiSeq 2500 system (Illumina, San Diego, CA) and the sequenced reads were mapped to an annotated human genome (version GRch37/hg19) using STAR (*Dobin et al., 2013*). The value of FPKM (Fragments Per Kilobase of transcript per Million mapped reads) calculated by Cufflink algorithm was used to represent the gene expression level (*Trapnell et al., 2010*). The RNAseq data (GSE75719) was submitted to NCBI GEO database.

## Lentivirus packaging, shRNA knockdown and stable transformation of HK2 and LDHA to NPCs

Lentiviral plasmids containing shRNA (Mission shRNA, Sigma) against human LDHA (TRCN0000026538; TRCN0000164922); HK2(TRCN0000195582; TRCN0000232926); PGC-1α (TRCN0000001169; TRCN0000001166), ERRγ (TRCN0000033645; TRCN0000033647) were transfected into 293T cells using Lipofectamine 2000 (Invitrogen) together with the third generation packaging plasmids pMD2.G, pRRE and pRSV/REV. Cells were cultured in DMEM containing 10% fetal bovine serum, 100 U/mL penicillin and 100 µg/mL streptomycin. Lentivirus was concentrated from filtered culture media (0.45 µm filters) by ultracentrifugation at 25,000 rpm for 90 min. After two days of infection, NPCs or neurons were selected with 200 ng/ml puromycin (Sigma). The inducible c-MYC (T58A, a mutation stabilizing c-MYC) lentivirus expression vectors, FUdeltaGW-rtTA (#19780) and FU-tet-o-hc-MYC (#19775) were developed by *Maherali et al. (2008)* and obtained from Addgene. Doxycycline (Sigma) was used at 2 µg/ml. HK2 and LDHA (Flag-tagged) were cloned into home-made piggyBac transposon-based vector. HK2 and LDHA are spaced by a self-splicing 2A peptide and their expression is controlled by CMV promoter. NPCs were transfected with high-efficiency Amaxa Nuclearfector technology (Lonza, Basel, Switzerland).

## GC-MS analysis of metabolites

Neurons were grown in a 6-well plate. After growth in fresh medium for 12 hr, cells were washed quickly 3 times with cold PBS, and 0.45 ml cold methanol (50% v/v in water with 20 µM L-norvaline as internal standard) was added to each well. Culture plates were transferred to dry ice for 30 min. After thawing on ice, the methanol extract was transferred to a microcentrifuge tube. Chloroform (0.225 ml) was added and the tubes were vortexed and centrifuged at 10,000 g for 5 min at 4°C. The upper layer was dried in a centrifugal evaporator and derivatized with 30 µl O-isobutylhydroxyl-amine hydrochloride (20 mg/ml in pyridine, TCI) for 20 min at 80°C, followed by 30 µl N-tert-butyldi-methylsilyl-N-methyltrifluoroacetamide (Sigma) for 60 min at 80°C. After cooling, the derivatization mixture was transferred to an autosampler vial for analysis. GC-MS analysis was performed in the Cancer Metabolism Core at the Sanford-Burnham Medical Research Institute (La Jolla, California). More details including the parameters of machine settings can be found in the publication from the center (*Scott et al., 2011*).

## OCR and lactate measurement

The OCR of NPCs and neurons grown in a laminin-coated Seahorse plate was measured using a Seahorse extracellular Flux Analyzer (Agilent Technologies Inc, La Jolla, CA), following the manufacturer's instructions. After the measurement, cells were lysed in 60~100 µl lysis buffer with two 'freeze and thaw' cycles on dry ice. Protein concentrations were determined by DC protein assay (Bio-Rad). The OCR values were normalized by protein mass. For measurement of lactate levels in medium, medium from cultures of iPSCs, NPCs and neurons was freshly changed and collected after 12 hr; cells were then frozen on the plate and lysed by two freeze-and-thaw cycles in dry ice. Medium lactate was measured by Lactate Assay kit (BioVision, Milpitas, CA) and normalized by total protein content.

## Statistical analysis

Comparisons were done using Student's t-test. Statistical analyses were performed using GraphPad Prism.

## Acknowledgements

We thank members of the Hunter lab for helpful discussions, and Jill Meisenhelder, Suzy Simon and Justin Zimmerman for laboratory support. This study was supported by NIH grants (CA14195, CA80100 and CA82683) to TH, and the G. Harold & Leila Y Mathers Charitable Foundation, the JPB Foundation, the Leona M. and Harry B Helmsley Charitable Trust grant #2012-PG-MED002, Annette Merle-Smith, CIRM (TR2-01778) to FHG. The study was also supported by Salk core facilities and their staff, including the Stem Cell Core, Advanced Biophotonics, Flow Cytometry, Bioinformatics, and Functional Genomics core. We are thankful for the support from the Helmsley Center for Genomic Medicine. TH is a Frank and Else Schilling American Cancer Society Professor and the Renato Dulbecco Chair in Cancer Research. XZ was supported by a fellowship from the California Institute of Regenerative Medicine and a Salk Pioneer postdoctoral fellowship.

## Additional information

### Funding

| Funder | Grant reference number | Author |
| --- | --- | --- |
| California Institute for Regenerative Medicine | | Xinde Zheng |
| Salk Pioneer postdoctoral fellowship | | Xinde Zheng |
| California Institute for Regenerative Medicine | TR2-01778 | Fred H Gage |
| The G. Harold and Leila Y. Mathers Foundation | | Fred H Gage |

| JPB Foundation | | Fred H Gage |
|---|---|---|
| Leona M. and Harry B. Helmsley Charitable Trust | #2012-PG-MED002 | Fred H Gage |
| National Institutes of Health | CA14195 | Tony Hunter |
| National Institutes of Health | CA80100 | Tony Hunter |
| National Institutes of Health | CA82683 | Tony Hunter |

The funders had no role in study design, data collection and interpretation, or the decision to submit the work for publication.

## Author contributions

XZ, Established neural progenitor cells, neuron differentiation and performed cell biology experiments, Performed RNA-seq, metabolic analysis and knockdown experiments, Performed data analysis and interpretation, Wrote the manuscript, Conception and design; LB, Established neural progenitor cells, neuron differentiation and performed cell biology experiments, Performed data analysis and interpretation, Wrote part of the manuscript, Conception and design; MJ, Established neural progenitor cells, neuron differentiation and performed cell biology experiments, Performed molecular biology experiments, Conception and design, Analysis and interpretation of data; JM, Provided reprogrammed iN, Acquisition of data, Analysis and interpretation of data; YK, Assisted mitochondria-related assay, Analysis and interpretation of data, Contributed unpublished essential data or reagents; LMa, MH, Assisted in lentivirus shRNA experiments, Analysis and interpretation of data; LMa, Assisted metabolic analysis, Analysis and interpretation of data, Contributed unpublished essential data or reagents; FHG, Conception and design, Drafting the article, Analysis and interpretation of data; TH, Conception and design, Analysis and interpretation of data, Drafting or revising the article

## Author ORCIDs

Tony Hunter, http://orcid.org/0000-0002-7691-6993

## Additional files

### Supplementary files

• Supplementary file 1. Real time PCR primers.

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
