## [Decision Letter]

Thank you for submitting your work entitled "Metabolic reprogramming during neuronal differentiation from aerobic glycolysis to neuronal oxidative phosphorylation" for consideration by *eLife*. Your article has been reviewed by two peer reviewers, including Andrew Pieper (Reviewer #1), and the evaluation has been overseen by a Reviewing Editor and a Senior Editor.

The reviewers have discussed the reviews with one another and the Reviewing Editor has drafted this decision to help you prepare a revised submission.

Summary:

This work sheds new light on alterations in metabolic gene expression during differentiation of neural progenitor cells (NPCs) to neurons. Authors observe suppressed glycolytic gene expression in neurons, along with reduced expression of the glycolytic regulators c-MYC and N-MYC. The promoter for lactate dehydrogenase-A (LDHA) becomes methylated during neuronal differentiation, preventing either hypoxia or MYC overexpression from rescuing LDHA expression in neurons. Enforcing constitutive expression of glycolytic genes impairs differentiation of NPCs into viable neurons.

Essential revisions:

While both reviewers found merit and interest in the work, both indicated that some translation to the in vivo situation is required. Additionally, Reviewer 2 has very cogent and important suggestions that must be taken into account in a revised manuscript. With respect to demethylation in neurons, it would be powerful to show somehow that reversing promoter methylation allowed neurons to become glycolytic again. This would be particularly valuable in the context of hypoxia because of the relevance to stroke. One possibility would be to try to suppress methylation in the proliferating NPCs using 5 aza during the differentiation assay, since it is not possible to cause demethylation with this agent in non-dividing neurons. If neuronal differentiation can still occur under those conditions (a big if), then the authors could take those neurons (presumably with unmethylated HREs in the LDHA promoter) subject them to hypoxia and test for induction of the LDHA mRNA. Obviously this treatment would alter the methylation status of many genes, so the authors may find that neuronal differentiation is impaired to the point that the experiment cannot be performed. We think authors should give this experiment serious thought – it should not be difficult and could be very informative.

[Editors' note: further revisions were recommended prior to publication, as described below.]

Congratulations: we are very pleased to inform you that your article, "Metabolic reprogramming during neuronal differentiation from aerobic glycolysis to neuronal oxidative phosphorylation", has been accepted for publication in *eLife*. The Reviewing Editor for your submission was Gail Mandel.

We include the re-review comments of Reviewer #2 below; unfortunately those comments were accidentally omitted from the original decision. If the authors have this data and can include it, it would make their case stronger.

*Reviewer #2:*

I recommend accepting the manuscript. It's an interesting story. I just have one hopefully simple suggestion. I realize that the authors did not get all the comments from the first round and therefore may have missed the suggestion to provide some direct evidence of increased glycolysis in the NPCs relative to differentiated neurons. I still think they should do that if possible; the evidence that oxygen consumption is enhanced during neuronal differentiation is quite strong, but the evidence that glycolytic activity goes down is somewhat weaker in my mind, and this might be easily rectified – it could be as simple as showing the extracellular acidification rate (ECAR) data from the Seahorse experiment in Figure 1—figure supplement 6. Including ECAR data would satisfy most metabolism readers. However, this can be considered as a discretionary change as the large amount of indirect evidence already in the paper supports the idea of glycolytic suppression during differentiation. I'd be happy to briefly review the data should the authors choose to include it.

---

## [Author Response]

*While both reviewers found merit and interest in the work, both indicated that some translation to the in vivo situation is required. Additionally, Reviewer 2 has very cogent and important suggestions that must be taken into account in a revised manuscript. With respect to demethylation in neurons, it would be powerful to show somehow that reversing promoter methylation allowed neurons to become glycolytic again. This would be particularly valuable in the context of hypoxia because of the relevance to stroke. One possibility would be to try to suppress methylation in the proliferating NPCs using 5 aza during the differentiation assay, since it is not possible to cause demethylation with this agent in non-dividing neurons. If neuronal differentiation can still occur under those conditions (a big if), then the authors could take those neurons (presumably with unmethylated HREs in the LDHA promoter) subject them to hypoxia and test for induction of the LDHA mRNA. Obviously this treatment would alter the methylation status of many genes, so the authors may find that neuronal differentiation is impaired to the point that the experiment cannot be performed. We think authors should give this experiment serious thought – it should not be difficult and could be very informative.*

We are very thankful for the reviewers’ suggestions. To extend this observation to the in vivo situation, we first set out to examine the *LDHA* promoter methylation in primary mouse (E17) and rat (E18) embryonic neurons, but, to our surprise, in contrast to human neurons differentiated from NPCs in culture, we were unable to detect *LDHA* promoter methylation. We also checked for LDHA promoter methylation in adult human brain tissue, but again no methylation was detected. Moreover, we could not detect *LDHA* promoter methylation in neurons differentiated from mouse NPC lines. However, we found that, as was the case for human neurons, even in the absence of promoter methylation, the expression of LDHA decreased greatly during mouse neuronal differentiation, as did mouse HK2 as seen in Figure 3—figure supplement 1. Clearly, *LDHA* promoter methylation is not essential for its decreased expression in differentiated neurons. The key mechanism underlying the metabolic switch occurring during neuronal differentiation, turning off LDHA and HK2 expression, is conserved between human and mouse.

Currently, we do not have an experimental explanation for why *LDHA* promoter is only methylated in human NPC-derived neurons but not mouse neurons. This issue will require more extensive investigations and efforts beyond the time allowed of current revision. Thus, we have decided to report the LDHA promoter methylation story separately when the mechanism becomes clear. In this new submission, part of the original Figure 4 and Figure 5, describing the effects of methylation on MYC and HIF1 binding, have been removed. The main mechanism of metabolic switch, turning off HK2 and LDHA, is not affected but strengthened by the newly added data from mouse neuronal differentiation. We believe that the rest of story still presents a novel view of metabolic switch during neuronal differentiation.

Although, this issue is now less relevant, we followed the reviewer's suggestion and tried to suppress de novo methylation during the neuronal differentiation using 5'-aza-2’-deoxycytidine (5-azaC). We found that the addition of 5-azaC greatly impaired neuronal differentiation and caused extensive cell death, a potential concern already raised by the reviewer. We did not observe such a "toxic" effect with 5-azaC treatment on already-differentiated 3-week neurons. The result indicates that methylation is critical for neuronal differentiation. We tested different concentrations of 5-azaC, and a representative result is shown in Figure 7.

Author response image 1.Neuronal differentiation in the presence of 5'-aza-2'-deoxycytidine (2 µM) or DMSO as control.NPC-derived neurons after 2 weeks of differentiation were stained of MAP2, a neuronal marker.**DOI:**
http://dx.doi.org/10.7554/eLife.13374.031

[Editors' note: further revisions were recommended prior to publication, as described below.]

Reviewer #2:

I recommend accepting the manuscript. It's an interesting story. I just have one hopefully simple suggestion. I realize that the authors did not get all the comments from the first round and therefore may have missed the suggestion to provide some direct evidence of increased glycolysis in the NPCs relative to differentiated neurons. I still think they should do that if possible; the evidence that oxygen consumption is enhanced during neuronal differentiation is quite strong, but the evidence that glycolytic activity goes down is somewhat weaker in my mind, and this might be easily rectified – it could be as simple as showing the extracellular acidification rate (ECAR) data from the Seahorse experiment in Figure 1—figure supplement 6. Including ECAR data would satisfy most metabolism readers. However, this can be considered as a discretionary change as the large amount of indirect evidence already in the paper supports the idea of glycolytic suppression during differentiation. I'd be happy to briefly review the data should the authors choose to include it.

We are very thankful for the reviewer's suggestion. The basal ECAR values of NPCs and differentiated neurons from the Seahorse experiment in Figure 1—figure supplement 6 are reproduced below in Figure 8. The basal ECAR of NPCs is ~2.5 fold of that of neurons, consistent with decreased glycolysis during neuronal differentiation. However, we believe that this 2.5 fold change in ECAR may underestimate the magnitude of the difference between NPCs and neurons for the following technical reason. For this experiment, NPCs had to be grown at high density to avoid differentiation, and the large amount of lactate secreted by high density NPCs under these conditions was sufficient to decrease the pH value of the medium, which lacks sodium bicarbonate, significantly, from 7.4 to 7.0, during incubation and measurement. This meant that the NPC ECAR readings were out of the linear measurement range based on our own experience, and also communications with Seahorse technical support staff. Thus, we decide not to put the data into the paper. To clarify the point, we add the following sentences in the figure legend of Figure 1—figure supplement 6: "In this experiment, NPCs had to be grown at high density to avoid differentiation and ensure optimal proliferation. The large amount of lactate secreted by high density NPCs under these conditions decrease the pH value of the medium, which lacks sodium bicarbonate, significantly, from 7.4 to 7.0, during incubation and measurement. The NPC ECAR readings were out of the linear measurement range and not shown here". We took the alternative approach of measuring the lactate/glucose production ratio, which is also a common way of measuring glycolytic activity in cells. As shown in Figure 1—figure supplement 7, the ratio was ~1.61 lactate molecules/glucose molecule for NPCs, whereas in neurons it was ~0.35, consistent with a significant decrease in aerobic glycolysis as NPCs differentiate into neurons. Consistently, the cellular level of lactate in neurons measured by GC-MS is less than 10% of that in NPCs (Figure 1—figure supplement 5). We hope that the reviewer finds this response satisfactory.

Author response image 2.Extracellular acidification rate (ECAR) by Seahorse extracellular flux analysis on NPC and 3-week neurons.Error bars represent SD.**DOI:**
http://dx.doi.org/10.7554/eLife.13374.032